# High Variability Periods in the EEG Distinguish Cognitive Brain States

**DOI:** 10.3390/brainsci13111528

**Published:** 2023-10-30

**Authors:** Dhanya Parameshwaran, Tara C. Thiagarajan

**Affiliations:** Sapien Labs, 1201 Wilson Blvd, 27th Floor, Arlington, VA 22209, USA

**Keywords:** electroencephalography (EEG), entropy, anesthesia, brain states, working memory

## Abstract

Objective: To describe a novel measure of EEG signal variability that distinguishes cognitive brain states. Method: We describe a novel characterization of amplitude variability in the EEG signal termed “High Variability Periods” or “HVPs”, defined as segments when the standard deviation of a moving window is continuously higher than the quartile cutoff. We characterize the parameter space of the metric in terms of window size, overlap, and threshold to suggest ideal parameter choice and compare its performance as a discriminator of brain state to alternate single channel measures of variability such as entropy, complexity, harmonic regression fit, and spectral measures. Results: We show that the average HVP duration provides a substantially distinct view of the signal relative to alternate metrics of variability and, when used in combination with these metrics, significantly enhances the ability to predict whether an individual has their eyes open or closed and is performing a working memory and Raven’s pattern completion task. In addition, HVPs disappear under anesthesia and do not reappear in early periods of recovery. Conclusions: HVP metrics enhance the discrimination of various brain states and are fast to estimate. Significance: HVP metrics can provide an additional view of signal variability that has potential clinical application in the rapid discrimination of brain states.

## 1. Introduction

The high temporal resolution of the EEG, as well as its easy portability relative to other neuroimaging methods such as fMRI, allows for both ease of use and greater insight into the temporal behavior of the brain. EEG, thus, has tremendous potential in the context of clinical applications that require the discrimination of neurological conditions and brain states. These include BCI applications, anesthesia monitoring, seizure detection, and diagnoses of cognitive and neurological disorders. In this context, metrics that can serve as biomarkers both individually and in combination with other metrics are of great interest.

Spatiotemporal approaches such as micro-state analysis that examines fast switching between neural assemblies in the resting brain state differ between various behavioral and task conditions [1,2,3]. Similarly, high amplitude bursts in EEG termed avalanches have been shown to be predictive of visual tasks and reported emotion [4], and entropy and connectivity measures have been combined to predict emotional state [5]. However, such spatiotemporal analysis typically requires large or high-density electrode configurations. Equally, various single-channel metrics carry significant information about brain states and behavior with the advantage of having applications in cost-effective and portable devices with lower electrode configurations. For instance, the strength of alpha oscillation increases when eyes are closed, discriminating between eyes’ open and closed states [6,7,8], and peak alpha frequency has been shown to correlate with cognitive traits [9] while other spectral measures have been shown to distinguish states of consciousness under anesthesia [10]. So also, various entropy measures applied to the EEG signal have been shown to change under disease conditions such as Alzheimer’s [11], concussion [12], and disorders of consciousness [13], as well as decrease under anesthesia [14,15,16,17] while EEG waveform complexity, a measure similar to entropy but focused on longer waveform patterns was correlated with performance on a pattern completion task [18]. However, while these various analytical approaches to the signal demonstrate differences between behaviors and brain states, there is still a substantial challenge in distinguishing brain states with the certainty required for clinical application. Thus, delivering on the potential of EEG depends on continued exploration of analytical approaches to the signal that can improve the prediction of brain states and conditions and serve as biomarkers for distinct neurological and mental health conditions. Furthermore, given that there are numerous behavioral conditions and brain states, individual metrics tend to map to multiple states and outcomes. Thus, multiple orthogonal perspectives of the signal are needed to provide more accurate and precise predictions of brain states and deeper insights into the relationship between brain states and outcomes.

There are numerous metrics in the literature that have been used to characterize the variability of the EEG signal, such as various entropy and complexity measures, as well as harmonic regression. However, each of these examines distinct aspects of the signal variability from spectral variability (e.g., spectral entropy) to amplitude variability (sample entropy) to waveform shape (waveform complexity) and deviation from a harmonic signal (harmonic regression). Embedded in these are implicit assumptions of which aspects of the signal are most important. For example, spectral entropy assumes that the frequency composition is most important, while waveform complexity assumes that the shape of the waveform, which maintains the relative phase relationships among frequencies, is more important. Sample entropy, on the other hand, utilizes a vector distance measure on short-time scales. However, these are not exhaustive in their characterization of signal variability. Furthermore, it is unclear which aspect of variability of this aggregate signal is most important in the context of task outcomes. Given this, various approaches must be tried, with each providing a different view and insight into the relationship between outcomes and underlying activity. 

In this paper, we describe a novel set of metrics that characterize high variability periods (HVPs) in the signal. In contrast to the other measures of variability, the metrics we propose look at the shifting between periods of low and high variability over time. Given that higher variability implies a greater degree of change and, therefore, energy of the signal, this approach conjectures that the way the shifts in the periods of high variability of the signal occur could carry meaningful information about brain state, the type of activity or task the brain is performing, and how well it is performing that task. We demonstrate the behavior of this metric with different parameter choices of the algorithm. In addition, as an initial demonstration of its potential, we show that this set of metrics shows significant differences between various conditions of cognitive task performance that outperform commonly used single channel metrics such as entropy, complexity, and harmonic regression and also decreases significantly under anesthesia. We further show that these metrics are not significantly correlated with these other measures, which make different assumptions on both timescales and the aspect of variability that is characterized. Further when combined with these traditional metrics in supervised learning models, they significantly enhance the ability to predict when an individual has their eyes open or closed or is performing a working memory task versus a pattern completion task. Thus, they are novel in the information they provide about the signal. Finally, these metrics are computationally faster to estimate than other single-channel metrics and easy to implement with the low-electrode configurations.

## 2. Methods

### 2.1. Human EEG Recordings

Our demonstration of the computation and properties of this metric utilizes EEG recordings obtained from adults between the ages of 21 and 50, as previously described in [19]. EEG recordings were obtained from 50 participants sitting quietly for three minutes with their eyes closed (EC) or with their eyes open and looking at images on a laptop screen (EO). For a subset of 28 participants, EEG was recorded performing a working memory (EO-WM) task of retracing a pattern on a grid or a pattern completion task (EO-PC) utilizing a raven’s progressive matrix. Each of these tasks was completed over a 2 to 2.5 min period. The recordings were obtained using the 14-channel Emotiv EPOC+ (channels AF3, F7, F3, FC5, T7, P7, O1, O2, P8, T8, FC6, F4, F8, AF4), which utilizes a bilateral mastoid reference (M1, a ground reference point for measuring the voltage of the other sensors and M2, a feed-forward reference point for reducing electrical interference from external sources). The EPOC+ signals were high-pass-filtered with a 0.16 Hz cutoff, pre-amplified, and low-pass-filtered at an 83 Hz cutoff. The analog signals were then digitized at 2048 Hz and filtered using a 5th-order sinc notch filter (50 and 60 Hz) before being down-sampled to 128 Hz (company communication). The effective bandwidth of the signal is, therefore, between 0.16 and 45 Hz. For certain analysis, subsets of frontal and temporal channels were used as follows: subset **F8** (AF3, AF4, F7, F8, FC5, FC6); **F4_1** (F3, F4, F7, F8); **F4_2** (AF3, AF4, F7, F8); **T4** (T7; T8, FC5, FC6). Unless otherwise indicated, metrics shown for each individual are averaged across channels.

### 2.2. Monkey EEG Recordings

EEG and ECOG simultaneous recordings from monkeys were obtained from Neurotycho, an open-access collection of multidimensional, invasive EEG recordings from multiple macaque monkeys [20,21]. The EEG signal was recorded in two monkeys from 19 channels with a sampling rate of 4096 Hz. The location of the EEG electrodes was determined by a 10–20 system without Cz using the NeuroPRAX system (neuroConn GmbH, Ilmenau, Germany). EEG signals were referenced to an average between the signals recorded from the bilateral mastoids. Recordings were down-sampled to 128 Hz to match the human recordings. 

For the anesthesia experiment, EEGs were recorded after the injection of an anesthetic agent and an antagonist in two monkeys. The monkey was seated in a primate chair with the movements of both arms and the head restricted, and heart rate and respiration were monitored during the entire experiment. The anesthetic agent, a combination of ketamine (8.21 mg/kg for M1 and 5.00 mg/kg for M2) and medetomidine (0.05 mg/kg for M1 and 0.02 mg/kg for M2), was injected at the same time intramuscularly. Dosage was titrated based on response. Data were collected in 5 min blocks. The depth of anesthesia was evaluated via pain stimulation reaction, and the low-anesthetic block started once they did not respond to it. The low-anesthetic block was immediately after injection, while the deep anesthetic block was recorded once sneeze, grasp, and blink reflexes had vanished and there was no response to pain stimulation. In the recovery state, 5 min of EEG was recorded shortly after the intramuscular injection of the antagonist (atipamezole, 0.21 mg/kg for M1 and 0.09 mg/kg for M2). 

### 2.3. Defining High Variability Periods (HVPs)

HVPs are identified by first computing the standard deviation of the amplitude values (A_SD) for a moving window of the signal of length m. Thus, for any window of length *m*
$$A\_SD=\sqrt{\overset{i+m.s}{\underset{i}{\sum }}\;\frac{{\left(A_{i}-\mu \left(A\right)\right)}^{2}}{m.s}}$$
where

*A_i_* = *i*th amplitude value; 

*μ*(*A*) = mean amplitude of the window;

*m* = length of window in seconds;

*s* = samples per second.

The average A_SD for different window sizes increased with window size up to 10 s but became relatively stable beyond 10 s (Figure 1A,D). Given that larger window sizes indicate larger samples of signal, and SD typically increases with sample size up to a point, this suggests that windows of 10 s provide a more stable estimation of SD. Figure 1B,E shows that the EEG signal (grey) normalized to the overall channel A_SD, as well as the A_SD values for three different values of *m* (5, 10, and 15 s), shifted with a 50% overlap for one human and one monkey, respectively. The amplitude value corresponding to the quartile value of the A_SD values computed across all windows across the full length of the signal is then used as a threshold T (dotted line in Figure 1B,E). The quartile value is defined as the *i*th value in the rank-ordered A_SD values, where *i* = 25n/100, and n is the number of points = L/*m*, where L is the length of the signal, and *m* is the window size.

High Variability Periods or HVPs are then defined as those stretches where consecutive A_SD values are continuously above the threshold T (Figure 1C,F), while low variability periods or LVPs are defined as those stretches where the A_SD values are continuously below the threshold *T* as follows: $$HVP=\left(i\left(T_{n+1}\right)-i\left(T_{n}\right)\right)\;\forall \;i\left(T_{n}\right)\;where\;A\_SD(i(T_{n})+1)>T$$
$$LVP=\left(i\left(T_{n+1}\right)-i\left(T_{n}\right)\right)\;\forall \;i\left(T_{n}\right)\;where\;A\_SD(i(T_{n})+1)<T$$
where 

*T_n_* = the nth threshold crossing;

*i* = the position in the sequence of *A_SD* values.

Figure 1B,E show the EEG signal normalized by A_SD value in order to accommodate both the signal and moving window on the same graph. Other figures utilize the amplitude SD value (in µV), corresponding to the SD of the EEG signal.

### 2.4. Metrics That Describe High Variability and Low Variability Periods (HVPs and LVPs)

We compute the following metrics: (1) The average HVP duration (HVP-D) defined as the average width of all HVPs across the recording; (2) The average HVP area (HVP-A) computed as the average of the sum of all values across the HVP periods; and (3) The LVP duration (LVP-D) defined as the average width of all LVPs across the recording as follows:$$\begin{array}{c} HVP_{-}D=m\left(i\left(T_{n+1}\right)-i\left(T_{n}\right)\right)\;\left(1\right)\;\forall i\left(T_{n}\right)\;where\;A_{-}SD\left(i\left(T_{n}\right)+1\right)>T\\ HVP_{-}A=\sum_{i\left(Tn\right)}^{i\left(Tn+1\right)}\;A_{-}SD\left(i\right)\;\left(2\right)\;\forall i\left(T_{n}\right)\;where\;A_{-}SD\left(i\left(T_{n}\right)+1\right)>T\\ LVP_{-}D=m\left(i\left(T_{n+1}\right)-i\left(T_{n}\right)\right)\;\;\;\;\;\;\;\left(3\right)\;\;\;\forall i\left(T_{n}\right)\;where\;A\_SD\left(i\left(T_{n}\right)+1\right)<T \end{array}$$

In addition, we also calculated the HVP rate (HVP-R) as the number of HVPs per minute of EEG recording and the ratio of the durations of HVPs to LVPs.

We note that to compare multiple conditions within the same recording (i.e., resting eyes closed and eyes open tasks), in order to have a consistent and comparable threshold in absolute amplitude units, the threshold was determined using the resting eyes closed (EC) in all conditions (Figures 2 and 3). In monkeys, when comparing multiple states in anesthesia (Figure 4), we used a 3 s window size to compute the A_SD. Furthermore, the quartile threshold T used to compute HVPs was determined based on the initial rest condition and applied to all subsequent states of low and deep anesthesia and recovery. Mean values were computed by first averaging across channels for each individual within a group and then computing the mean and standard error of these averages across all individuals.

### 2.5. Comparisons between Conditions

Metrics were compared between task conditions in human participants using Tukey’s honest significant difference (HSD) for analyzing the group differences (Appendix A). Further, to evaluate differences in metrics between conditions across individuals, we used a pairwise *t*-test.

### 2.6. Computing of Entropy and Waveform Complexity Metrics

Sample entropy, Lempel–Ziv entropy, and waveform complexity were computed as described in [18]. We also fit a harmonic regression to EEG segments of signal length *m* for a moving window along the signal duration with a 50% overlap, as performed for the HVP metric computation [22]. We performed a non-linear least squares fit using the equation EEG = *a* × sin (*bt* + *c*); the regression coefficient *a* represents the amplitude of the harmonic regression (HR) fit. To make a direct comparison to the HVP duration metric, we similarly used a quartile threshold for *a* to compute the average HR duration above the threshold.

### 2.7. Predicting Brain States

To determine how well HVP and other metrics were able to discriminate between each pair of tasks, we used the R implementation of extreme gradient boosting (xgboost version 2.0.0). Gradient boosting is a technique used to produce ensembles of decision trees incrementally by optimizing a loss function. We achieved it by using the 3 HVP metrics as well as sample entropy, harmonic metric duration, and spectral metrics, including the relative frequency component of Alpha, Beta, Gamma, Delta, and Theta bands, as well as the Theta/Beta ratio. Given the sample size of 28 individuals, 75% of the data was used to train the model, and the remaining 25% was used to test the model. For each of the metric combinations, we trained an xgboost model to a maximum depth of 3 and limited the boosting iterations to 50. We then built 10 models for different sets of training and testing data and computed model performance metrics of accuracy and F1 scores. Multiple performance measures are reported, including accuracy, sensitivity, specificity, recall, and F1 scores.

## 3. Results

### 3.1. Parameter Choice and HVP and LVP Characteristics in Humans

We first looked at how the durations of HVPs and LVPs varied as a function of window size in the eyes closed condition in the recordings from humans (Figure 2A,B; N = 50). On average, both HVP and LVP durations (HVP-D and LVP-D) were on the order of tens of seconds and increased sub-linearly with the window size. Figure 2A shows that average HVP and LVP durations increased for all window sizes when the window overlap was decreased from 50% to 10% while keeping the threshold constant as the mean value of the 25th percentile A_SD. Conversely, Figure 2B maintains the overlap at 50% and shows that HVP and LVP durations decrease for all window sizes as the threshold is increased from the 10th to 50th percentile of the A_SD. The ratio of HVP to LVP duration in the eyes closed condition remained relatively constant across all window sizes irrespective of the choice of window overlap but increased with decreasing threshold (Figure 2C). While the ratio was reasonably stable across window sizes for thresholds of the 25th and 50th percentiles, at a 10th percentile threshold, the HVP-D/LVP-D ratio became highly variable and decreased with window size (Figure 2D). This is most likely because the lower 10th percentile threshold may include substantial noise above the threshold. Thus, we recommend a quartile (25th percentile) threshold to eliminate noise but preserve much of the variability. Finally, as expected, both HVP and LVP rates decreased with window size (not shown in Figure 2). Values for all parameter choices for all computed metrics are shown in Appendix A.

### 3.2. Differences in HVP Characteristics between Behavioral States

Window size relates to the timescale on which the variability is meaningful, which, in turn, relates to various factors, such as the nature of the task and noise. Smaller window sizes will have fewer points and be more subject to noise, while longer window sizes may converge to a global average and no longer be meaningful. To identify an appropriate range of window size that could discriminate between conditions, we looked at mean HVP duration (HVP-D) across individuals as a function of window size *m* for each of four tasks: eyes closed (EC); eyes open passive (EO); working memory (EO-WM); and pattern recognition (EO-PR) (Figure 3A). HVP-D was significantly lower in the EC condition than EO and EO-PC across all window sizes and significantly higher than EO-WM (pairwise *t*-test for all comparisons for *m* = 10 and 25 s, Table 1, Appendix A). Among the EO tasks, HVP-D for EO-PC and EO were significantly higher than EO-WM (13.2 and 13.7 s at *m* = 10 s and 24.7 and 24.8 s at *m* = 25 s, respectively; *p* < 0.001 in both cases). HVP Area (HVP-A) was also similarly different across conditions, with greater differences between EO and EO-PC (Figure 3B). We also note that variability across individuals was greatest for the EC condition and decreased in EO and EO tasks. Overall, HVP-D in eyes open (EO) and pattern completion (EO-PC) were about twice as long as in the eyes closed (EC) condition for all window sizes between 10 and 25 s (Figure 3A; mean ± SEM EO = 73 ± 7 s, PC = 65 ± 5 s, EO-PC = 8, EC= 43 ± 1 s, and for *m* = 21 s) (Table 1). In the case of HVP-A, the pattern was distinct with EO 1.7–2.8-fold greater than all other conditions (Figure 3B, EO = 8.5 ± 2.4 mV.s, EO-PC = 3.6 mV.s, EO-WM = 5.5 mV.s, EC = 3.6 ± 0.5 mV.s for *m* = 21 s). Finally, the HVP-D/LVP-D ratio decreased dramatically with window size for all EO conditions in contrast to the EC condition, where it remained relatively more flat (Figure 3C). Altogether, we suggest that window sizes between 10 and 25 s can deliver good results, although the upper end may perform better.

### 3.3. Relationship to Other Temporal Metrics

The novel utility of HVPs depends on the metrics providing substantially different information about the signal relative to these other metrics. Here, we compared the HVP metrics to four other metrics of variability (harmonic regression (duration), sample entropy, Lempel–Ziv complexity, and waveform complexity; comparison to HVP duration for *m* = 10 s is shown in Table 2, and multiple window sizes are shown in Appendix A) to determine if they were correlated and, therefore, moved together. High correlations would indicate that they generally capture similar information about the signal. Harmonic regression looks at the fit and deviation of the signal from a harmonic function. However, it is distinct in its assumption of a symmetrical harmonic process with the same rate of an increase and decrease from nadir to peak and vice versa [12]. However, this is substantially at odds with the EEG, which is typically not a stable oscillatory process. Harmonic regression duration (the length of deviation from the harmonic function; see Methods) had the highest correlation to HVP-D but was, nonetheless, poorly correlated to HVP-D (r = 0.18, *p* = 0.37). Furthermore, harmonic regression metrics were not significantly different between brain states (Appendix A). 

Lempel–Ziv complexity, which measures repetitiveness in sequences, was most uncorrelated to HVP-D (r = 0.01, *p* = 0.97). Sample entropy, which quantifies the uncertainty within the signal, had a correlation to HVP-D of −0.09 (*p* = 0.68), while waveform complexity, which quantifies the diversity of waveform shapes in the signal, had a correlation value of 0.17 (*p* = 0.42). We note that these correlations are computed for a window size of 10 s; however, correlations are similarly low at all window sizes from 3 to 30 s (not shown). Thus, HVP metrics provide a substantially distinct view of the signal relative to these other metrics. 

Further, we also compared the computation time for each metric. HVP metrics and waveform complexity were orders of magnitude faster to compute for the same length of signal (0.1 s per channel for HVP metrics) compared to sample entropy and harmonic regression (7.4 and 53.7 s per channel; Appendix A).

### 3.4. Enhancing the Prediction of Brain States with HVP Metrics

Here, we use a supervised learning model (gradient boosting) to discriminate between each pair of the four conditions—eyes open (EO, no task), eyes closed (EC, no task), working memory (EO-WM), and pattern completion (EO-PC) (Table 3; all performance metrics in Appendix A). We show that HVP metrics on their own have accuracy and F1 scores of 0.88 and 0.86, respectively, when discriminating between eyes open and eyes closed compared to only 0.61 and 0.6 for the other entropy/harmonic metrics and 0.67 and 0.62 for spectral metrics. Combining all metrics did not improve performance relative to HVP metrics alone. On the other hand, in the cases of discriminating between EO-WM and EO and between EO-WM and EO-PC, HVP metrics alone performed similarly to all other metrics together and substantially enhanced accuracy and F1 scores by ~0.1 when considered in combination with other metrics. Thus, HVP metrics can substantially enhance predictive discrimination between certain brain states.

We further looked at the performance of HVP metrics in distinguishing between these conditions for different subsets of electrodes (Table 4). This included eight frontal electrodes (F8), two combinations of four frontal electrodes (F4_1 and F4_2), and four temporal electrodes (T4). F8 performed as well as the full set of 14 electrodes in discriminating between EO and EC, while F4_1 and T4 performed worse but still had reasonable discriminatory power. Both F8 and F4_2 performed as well as all channels together in distinguishing between EO-WM and EO, while frontal subsets performed better than all channels and T4 in discriminating between EO-WM and EO-PC.

### 3.5. HVP Metrics in Monkeys and the Effect of Anesthesia 

The HVP metrics in blindfolded monkeys followed a similar pattern to the eyes closed human condition (Figure 4A–C). Overall, HVP durations were slightly lower or similar at all window sizes for these blindfolded monkeys compared to the eyes closed state in humans (Figure 4A). While this suggests the possibility of interesting inter-species differences since the number of monkeys was only two and recording devices were different, it was not possible to make a statistical or reliable comparison between monkeys and humans. For an inter-species comparison, more directly comparable methodologies would need to be employed with a larger number of monkeys to draw reliable conclusions.

We next looked at how HVPs behaved under ketamine anesthesia in each of the two monkeys in recordings of 5 min blocks before anesthesia (rest), immediately after (two successive blocks called low and deep anesthesia), and after the injection of a reversing antagonist (recovery) (Figure 4D,E). Since data segments were short, we used a 3 s window. We showed that from the rest block to the low-anesthetic block, HVP duration decreased 2.75-fold from 7.6 to 2.8 s; HVP area decreased 5.2-fold from 1984 to 390 µVs, and HVP rate reduced 3.5-fold from 6/minute during the rest block to 1.7 HVPs/minute during the low-anesthetic block. HVPs then completely disappeared in the deep-anesthetic block and did not reappear in the initial recovery phase. (Figure 4F–H; Appendix A).

## 4. Discussion

### 4.1. Characteristics and Parameter Choice of the HVP Metric

Here, we describe a novel measure that provides a view of changes in temporal variability by characterizing periods of high amplitude variability in the signal. This metric is computationally light, is computed at the level of a single channel, and takes advantage of the high temporal resolution of the signal without making any assumptions about the underlying source or otherwise transforming the signal. Rather, it is simply a readout of shifts in temporal amplitude variability. The most important parameter is the threshold of A_SD used for the definition of the HVP metrics, as it has the greatest impact on outcome. The 25th percentile is recommended as a threshold that captures most of the signal variability but eliminates the variability of signal noise that contributes substantially to the 10th percentile. For window size, it is recommended to utilize both 10 s and 20 s, which can be compared, as they each have different HVP-D/LVP-D ratios for EO conditions relative to EC. Finally, the choice of overlap is less important in terms of outcomes. However, we recommend 50%, which is consistent with the default overlap used in many window-based algorithms from computations of the power spectrum to entropy.

Finally, it is of paramount importance that the threshold used to define HVPs is consistent across tasks and conditions in terms of amplitude units (µV) so that it is a like-to-like comparison of the two states. This is because computing thresholds separately for each condition, such as the eyes open and eyes closed, would result in different values since the A_SD value would be much higher in the eyes open condition. We, thus, recommend computing the threshold for the resting eyes closed state as a baseline and using this value in µV for all other tasks or conditions in the same individual. 

For all HVP metrics computed as we describe, we demonstrate that they are uncorrelated or only weakly correlated to other measures of variability, such as entropy, complexity and harmonic regression metrics, and, therefore, are not equivalent measures.

### 4.2. Utility of the Metric in Distinguishing Brain States or Behavioral Conditions

Here, we have provided several examples of the utility of the HVP metrics in distinguishing brain states and/or behavioral conditions. We have demonstrated that HVP metrics differ significantly between the conditions of eyes closed and eyes open, as well as between the performance of working memory and pattern recognition tasks in the eyes open state. Surprisingly, HVPs in the eyes open working memory task differ substantially from the eyes open passive condition but not eyes closed. This offers evidence that the differences are not substantially contributed by the eye blink that is present when eyes are open. Furthermore, distinct magnitude and significance of difference were found between conditions for different views of HVP (e.g., duration, area, HVP-D/LVP-D ratio), which suggests that together, they could provide a more accurate and precise distinction between task conditions than any individual metric alone. Going further, we have also demonstrated that the addition of HVP metrics to other traditional single-channel metrics in supervised learning models substantially enhanced the model performance in discriminating between the working memory, pattern completion, and eyes open task conditions.

We have also shown in monkeys that HVPs are suppressed by anesthesia and do not recover quickly with the application of the antagonist. The lack of recovery immediately after removal of the anesthetic suggests that it might reflect a cognitive state rather than consciousness per se and might be investigated in the context of conditions, such as postoperative cognitive impairment, which has been commonly described [23]. However, a lack of complete reversal of the effects of ketamine by atipamezole cannot be ruled out. Numerous other metrics have been explored to determine the depth of anesthesia, including cross-frequency coupling between alpha amplitudes in delta phases [24] and sample entropy [20,21]. Commonly used today is the BiSpectral Index (BIS), which is a proprietary combination of multiple metrics [25]. However, BIS is unreliable with several anesthetic drugs, including ketamine (the anesthetic used in the example we describe above), as well as other anesthetic drugs, and under certain neurological conditions [25,26]. This leaves open the possibility that greater accuracy could be achieved in monitoring the depth of anesthesia using HVP metrics in conjunction with other metrics. 

Altogether, these results suggest that HVP metrics provide a novel perspective of the signal that could have broad utility in enhancing the ability to distinguish brain states, particularly when used in conjunction with other orthogonal metrics.

### 4.3. Possible Applications of HVP Metrics

There are various potential applications of HVP metrics that can be explored. Given the ability to distinguish cognitive task conditions, a natural extension would be to determine if HVP metrics alone or in combination with other EEG metrics are able to better predict cognitive task performance. This would lend to applications in the diagnosis of cognitive dysfunction in development and aging. However, applications are not limited to cognition and may extend to discrimination of other brain states, such as depression or other psychiatric conditions, conditions such as fatigue versus alertness, where the ability to maintain high variability states for long periods or shift quickly between high and low states may be relevant.

The clear change in HVP metrics with anesthesia also suggests the potential for HVP metrics to contribute to various applications of real-time neurological monitoring, such as sleep and detection of seizures. In seizures, for instance, where behaviors become more stereotypical or even frozen, one hypothesis that could be explored is that the variability of the signal decreases, leading to longer low-variability periods. Similarly, different stages of sleep have been difficult to distinguish and may also exhibit changes in periods of variability that can be better discriminated with HVP metrics. Clinical application typically requires a high level of accuracy with low false positive and false negative rates, and, therefore, if HVP metrics can improve these parameters, they can have tremendous value. The computationally light characteristic of HVP metrics and the ability to compute them on relatively short time scales can also help enhance the utility of clinical applications that require real-time monitoring.

## 5. Conclusions

HVP metrics capture shifts in the variability of the EEG signal amplitude over window sizes of a few seconds. These measures capture a different dimension of the signal compared to complexity, entropy, and spectral measures of variability. In addition, they aid in the classification of various cognitive states. HVP metrics is, thus, an additional tool for EEG analysis that could enhance performance in a range of scientific and clinical applications.

## Figures and Tables

**Figure 1 brainsci-13-01528-f001:**
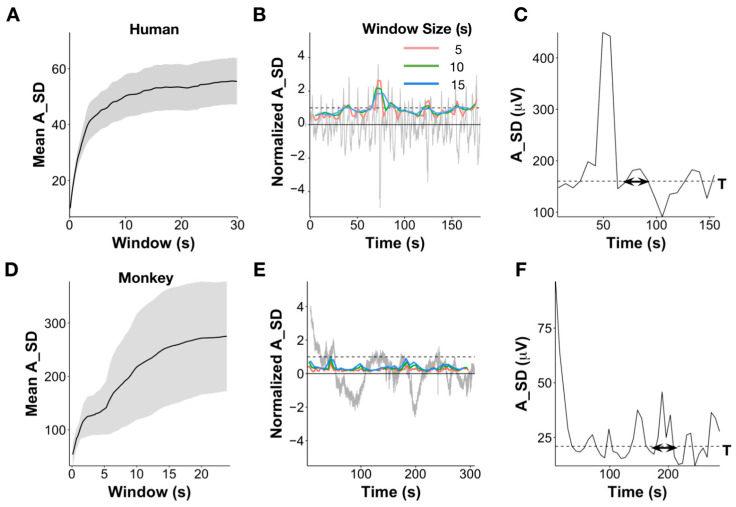
Definition of High Variability Periods (HVP). (**A**) Mean standard deviation of amplitude (A_SD in µV) across individuals for windows of different sizes. Error bars indicate standard error across individuals. (**B**) Resting state EEG signal from one person with amplitude shown in multiples of the standard deviation (A_SD, grey) along with the SD of the signal amplitude estimated for varying window sizes from 5, 10, and 15 s (colored lines). Dashed line represents the SD of the signal amplitude of the whole trace, i.e., A_SD or 1 SD of the amplitude. (**C**) A_SD using 15 s window and 50% overlap for the EEG signal, as shown in (**A**). The dashed line represents the 25th percentile of the moving A_SD values, which serves as the threshold T. All segments >T are defined as High Variability Periods or HVPs. Segments consistently <T are defined as low variability periods or LVPs. Durations are computed as the time between the threshold crossings. Area (for HVPs only) is computed as the sum of all SD values across the duration. (**D**) Mean standard deviation of amplitude (A_SD in µV) across 2 monkeys and 4 recordings for windows of different sizes. Error bars indicate standard deviation across monkeys. (**E**) Similar resting state signal along with SD estimates for different moving window sizes as shown in (**A**), but from a monkey rather than human. (**F**) The moving A_SD with a 15 s window and 50% overlap for the EEG signal, as shown in (**C**).

**Figure 2 brainsci-13-01528-f002:**
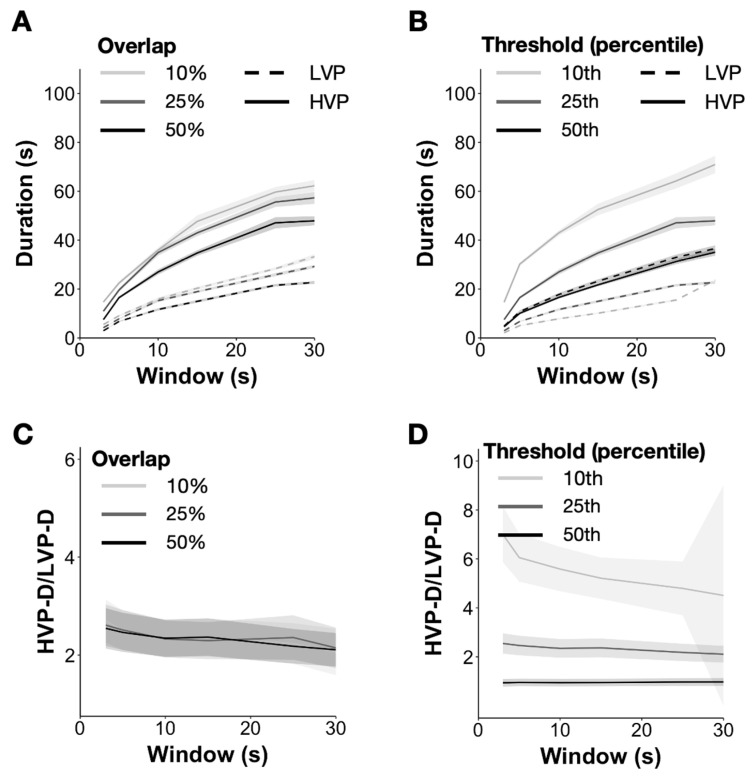
Parameter choices and HVP metrics for resting state eyes closed (EC). (**A**) Mean HVP and LVP durations across individuals for different window sizes (x-axis) and window overlaps (see legend) when threshold is maintained at the 25th percentile of the A_SD trace. (**B**) Mean HVP and LVP durations across individuals for different window sizes (x-axis) and thresholds (see legend), when window overlaps, is maintained at 50%. (**C**) Mean ratio of HVP and LVP durations across individuals for different window sizes (x-axis) and window overlaps (see legend) when threshold is maintained at the 25th percentile of the A_SD trace. (**D**) Mean ratio of HVP and LVP durations across individuals for different window sizes (x-axis) and thresholds (see legend) when window overlaps are maintained at 50%.

**Figure 3 brainsci-13-01528-f003:**
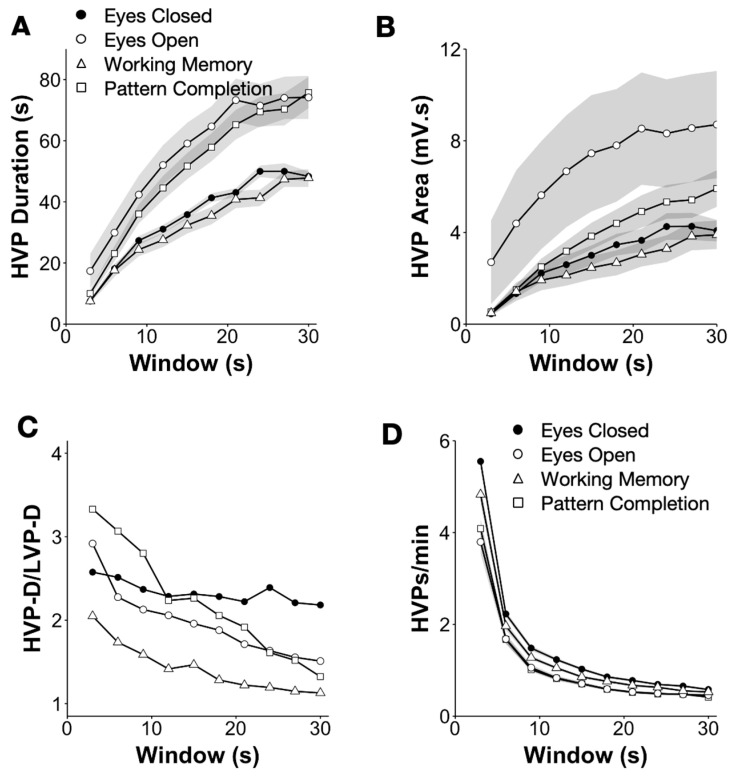
HVP metrics across behavioral conditions. (**A**) HVP duration (HVP-D; window overlap of 50% and 25th threshold percentile) across all 28 individuals (mean ± SEM) as a function of window size for resting eyes closed (EC; closed circle) and eyes open (EO; open circle) and for working memory (EO-WM) and pattern completion (EO-PC) tasks. (**B**) HVP area (HVP-A; mean ± SEM) as a function of window size for the same tasks/conditions as in (**A**). (**C**) Mean ratio of HVP and LVP duration as a function of window size for the same tasks/conditions as in A. (Error bars not shown for readability. SEM values are ~±0.2 to ±0.6). (**D**) HVP rate (mean ± SEM) as a function of window size for the same tasks/conditions as in (**A**).

**Figure 4 brainsci-13-01528-f004:**
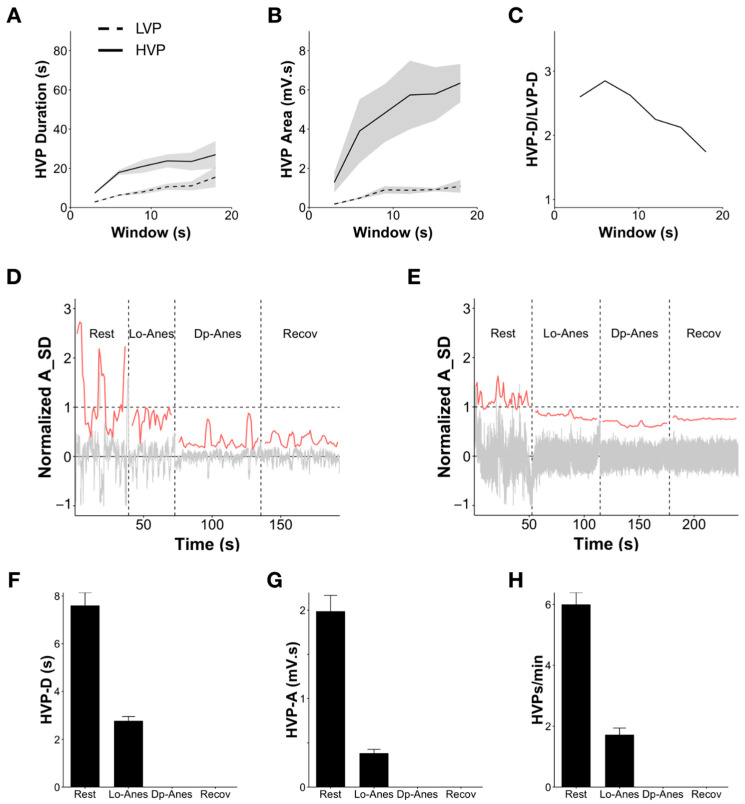
HVPs in monkeys and impact of anesthesia. (**A**) Mean HVP duration across 4 recording sessions across 2 blindfolded monkeys as a function of window size (overlap of 50% and threshold at 25th percentile). (**B**) Mean HVP area as a function of window size (overlap of 50% and threshold at 25th percentile). (**C**) Mean HVP-D/LVP-D ratio as a function of window size (overlap of 50% and threshold at 25th percentile). (**D**,**E**) Examples from a single channel in each of two monkeys showing the EEG signal (gray, normalized by the SD) before (rest) and after injection of ketamine (low-anesthetic and deep-anesthetic) and following injection of an antagonist for reversal of the anesthetic effect (recovery). The moving A_SD used to calculate HVPs is shown in red. The recording in (**A**) used a resting state threshold of 280 µV, and the recording in (**B**) used a resting state threshold of 153 µV. (**F**–**H**) HVP duration, area, and rate per minute in each state of anesthesia. HVP events completely disappeared in deep anesthesia and did not reappear in the initial recovery phase.

**Table 1 brainsci-13-01528-t001:** Differences in HVP metrics between eyes closed and eyes open task behaviors.

Condition 1	Condition 2	HVP-Duration	*p*-Value	HVP-Duration Diff (*m* = 25 s)	*p*-Value
		Diff (m = 10 s)			
Eyes Closed	Eyes Open	10.4	5.40 × 10^−2^	19.5 *	3.00 × 10^−2^
	Pattern Completion	9.9 *	3.10 × 10^−2^	19.6 *	1.00 × 10^−2^
	Working Memory	−3.3	2.10 × 10^−1^	−5.2	1.90 × 10^−1^
Eyes Open	Pattern Completion	−0.5	9.00 × 10^−1^	0.1	6.80 × 10^−1^
	Working Memory	−13.7 *	4.00 × 10^−3^	−24.7 **	0.00
Pattern Completion	Working Memory	−13.2 **	1.00 × 10^−3^	−24.8 **	0.00

Superscript * represents *p*-value less than 0.05, ** represents p-value less than 0.001.

**Table 2 brainsci-13-01528-t002:** Correlations between HVP Duration and alternate temporal measures.

Measure	Pearson’s r	*p*-Value
Harmonic Regression Duration	0.18	0.37
Lempel–Ziv Complexity	0.01	0.97
Sample Entropy	−0.09	0.68
Waveform Complexity	0.17	0.42

**Table 3 brainsci-13-01528-t003:** Prediction of brain states with HVP metrics alone and in combination with other metrics.

**EC vs. EO**		
	**Accuracy**	**F1-Score**
HVP metrics	0.88	0.86
Other Metrics	0.61	0.61
Other metrics + HVP-D	0.84	0.84
Other metrics + all HVP metrics	0.87	0.86
Spectral metrics	0.67	0.62
Spectral metrics + all HVP metrics	0.83	0.82
**WM vs. EO**		
	**Accuracy**	**F1-score**
HVP metrics	0.76	0.73
Other Metrics	0.75	0.73
Other metrics + HVP-D	0.82	0.81
Other metrics + all HVP metrics	0.85	0.83
Spectral metrics	0.77	0.72
Spectral metrics + all HVP metrics	0.73	0.71
**WM vs. PC**		
	**Accuracy**	**F1-score**
HVP metrics	0.63	0.59
Other Metrics	0.65	0.62
Other metrics + HVP-D	0.70	0.68
Other metrics + all HVP metrics	0.76	0.73
Spectral metrics	0.72	0.69
Spectral metrics + all HVP metrics	0.82	0.80

**Table 4 brainsci-13-01528-t004:** Prediction of brain states with HVP metrics for subsets of electrodes.

	EC vs. EO		EO vs. EO-WM		EO-WM vs. EO-PC	
Electrodes	Accuracy	F1-Score	Accuracy	F1-Score	Accuracy	F1-Score
ALL	0.88	0.86	0.76	0.73	0.63	0.59
F8	0.88	0.87	0.74	0.66	0.69	0.68
F4_1	0.65	0.66	0.63	0.61	0.67	0.66
F4_2	0.78	0.78	0.76	0.74	0.67	0.65
T4	0.81	0.79	0.67	0.58	0.64	0.59

## Data Availability

The data presented in this study from humans will be made available by the authors on request. The data presented in this study from monkeys are openly available on Neurotycho, http://www.neurotycho.org/eeg-and-ecog-simultaneous-recording (accessed on 24 October 2023).

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
