# Peer review of "High Variability Periods in the EEG Distinguish Cognitive Brain States"

_brainsci, 2023, doi:10.3390/brainsci13111528_

Round 1

Reviewer 1 Report

Comments and Suggestions for Authors

The paper presents a novel metric for assessing EEG signal variability, referred to as "High Variability Periods" (HVPs). These HVPs are defined as segments of the EEG signal characterized by a notably high standard deviation of amplitude. The study integrates HVP metrics with other EEG features, demonstrating their effectiveness in distinguishing various cognitive brain states, including those during wakefulness (eyes open and eyes closed), and their responsiveness to alterations in cognitive states induced by anesthesia. This research proposes that HVP metrics offer a swift and innovative approach to characterizing EEG signals and predicting changes in brain states.

I would like to raise a few points, addressing which would make the technique introduced in this work more applicable:

1. On the y-axis label of Figure 2 A and B, please remove "HVP" and retain only "duration," as it encompasses both HVP and LVP durations.

2. In Section 3.1, the conclusion should provide clear guidance for choosing parameters for the effective utilization of test cases.

3. Please ensure that the unit for HVP area is consistently included where necessary.

4. It appears that the S.E.M bars/shading is missing in Figure 3 C and D. Please rectify this issue.

5. The discussion should include guidance for potential uses of this metric in other cases, not shown here and also discuss the probable scientific reasoning behind the usefulness of this.

5. Kindly provide the appropriate names or details in the Author Contribution and Institutional Review Board sections as needed.

Author Response

We thank the reviewers for their careful and thoughtful review that has helped us substantially improve the manuscript.  We believe we have addressed all the points raised which includes the following major changes:

  1. New analysis showing task classification results for subsets of frontal and temporal electrodes (new Table 4)
  2. New analysis showing task classification results for spectral metrics (Table 3, Supp. Table 6)
  3. An additional section (4.4) has been added to discuss potential applications.
  4. Updates to labeling of Figure 2,3
  5. A rewrite of section 3.3 for greater clarity
  6. Addition of more mathematical descriptions in the methods and and numerical descriptions in various sections
  7. Addition of a conclusions section
  8. Addition of author contributions
  9. Addition of IRB information (moved from the results to end section)

Responses to the specific points attached in the PDF below. 

Reviewer 2 Report

Comments and Suggestions for Authors

1- Study design could be improved by comparing more complicated brain states than eye-closed, eye-open and .... A simple threshold on the signals of frontal electrodes could distinguish between such so-called brain states. 

2- It seems, applying conventional frequency-band dependent features (delta, theta, alfa and beta) are more suitable features for distinguishing such brain states. Add such comparisons to your reports.

Author Response

(The authors gave the same response as above.)

Reviewer 3 Report

Comments and Suggestions for Authors

As shown in the file

Author Response

(The authors gave the same response as above.)

Reviewer 4 Report

Comments and Suggestions for Authors

In this study, metrics derived from EEG signals were developed using different window sizes and overlap amounts. The contribution of these metrics to the classification performance of single-channel EEG signals recorded in four different conditions (EO-no task, EC-no task, EO-WM, and EO-PC) was examined. The study is generally well-founded; however, the following points need to be considered:

  1. 1- The motivation and novality of the study should be emphasized a bit more in the introduction section by referring to the features extracted from EEG signals present in the literature.
  2. 2- The study did not provide a detailed analysis of the effects of the proposed metrics on EEG channels and conditions. More information should be provided regarding which EEG channels perform better under which conditions and which EEG channels are sufficient for recording.

  3. 3- During the application of supervised learning models, the selection of training and testing data and whether methods such as k-fold cross-validation were used should be explained in more detail.

  4. 4- An evaluation of the recording duration of EEG signals should be conducted. It should be determined which time intervals are most meaningful for the computation of these metrics. For example, the impact of different recording durations such as 5 minutes or 1 minute on the classification performance of the metrics should be assessed.

Author Response

(The authors gave the same response as above.)

Round 2

Reviewer 2 Report

Comments and Suggestions for Authors

Accept after minor revision (conclusion part should be extended and explained more)

Author Response

We thank the reviewers for their thoughtful comments that have helped us bring more conceptual clarity to the novelty and utility of the HVP metrics. We have modified and extended the Introduction as well as the Discussion (sections 4.3 - 4.5) to address the points raised.

Reviewer2:

Accept after minor revision (conclusion part should be extended and explained more)

We have now extended the conclusion section as follows:

Line 546, Section 4.5

HVP metrics capture shifts in variability of the EEG signal amplitude over window sizes of a few seconds. These measures capture a different dimension of the signal compared to various complexity, entropy and spectral measures of variability. In addition, they aid the classification of various cognitive states. HVP metrics are an additional tool for EEG analysis that could enhance performance in a range of scientific and clinical applications.

Reviewer 3 Report

Comments and Suggestions for Authors

Why does the metric capture something that other more broadly used metrics don’t? The HVP metric needs to be more clearly defined and explained. Actually, the writeup lacks theoretical foundation – explanation. While the paper does show evidence as to the metric being different from the other metrics, it is not clear fundamentally / conceptually why they are different.

We have now added a paragraph in the introduction to makes this clear as follows:

There are various approaches and metrics in the literature that have been used to

characterize the variability of the signal such as various entropy and complexity measures as well as, harmonic regression. However each of these examine distinct aspects of the signal variability from spectral variability (e.g. Spectral entropy) to signal uncertainty in the time domain (Sample entropy) to waveform shape (waveform complexity), deviation from a harmonic signal (harmonic regression). Embedded in these are also assumptions about timescales or frequencies.

However these are not exhaustive in their characterization of variability.

Unfortunately, the authors do not answer the question “it is not clear fundamentally / conceptually why they are different.”  They do not contrast strengths of their approach (only what other approaches do).

2. Why was the protocol information on p. 3 removed?

3. Argument is incomplete:

anesthesia in monkeys. Estimating depth of anesthesia is a major clinical application and while bispectral index and entropy are both used in clinical settings for this purpose there is scope to improve this estimation and HVP metrics could be assessed in this context.

ð  How does this improve? Not clarified.

Another majorpotential clinical application is in

=> Nothing given here.

We have now added a section in the discussion on scientific and clinical applicability as follows:

=> Nothing given here.

4.4 You speculate (below) without at least some hypothesis for each.

The clear change in HVP metrics with anesthesia also suggests potential for HVP metrics to contribute to various applications in real-time neurological monitoring such as in detection of seizures, monitoring anesthesia and sleep. Clinical application typically requires a high level of accuracy with low false positive and false negative rates and therefore metrics that can improve these parameters in clinical practice can have tremendous value. For real time monitoring metrics that are computationally light and can be computed on short time scales like the HVP metrics are particularly useful.

3. Fig. 1: “ Table 15. second window” This is not clear.

4. p. 18 – Justify “However, applications are not limited 518 to cognition and may extend to discrimination of other brain states such as depression or other psychiatric conditions or symptoms or conditions such as fatigue versus alertness.”  Right now it is pure speculation.

Author Response

We thank you for the thoughtful comments that have helped us bring more conceptual clarity to the novelty and utility of the HVP metrics. We have modified and extended the Introduction as well as the Discussion (sections 4.3 - 4.5) to address the points raised.

Point by Point response in the attached file

Reviewer 4 Report

Comments and Suggestions for Authors

The article has provided explanations on the topics I mentioned. The article can be accepted in its current form.

Author Response

We thank you for the thoughtful comments that have helped us bring more conceptual clarity to the novelty and utility of the HVP metrics. We have modified and extended the Introduction as well as the Discussion (sections 4.3 - 4.5) to address the points raised by two other reviewers.